# The Importance of Correlation between CBCT Analysis of Bone Density and Primary Stability When Choosing the Design of Dental Implants—Ex Vivo Study

Mirko Mikic [1], Zoran Vlahovic [2], Momir Stevanović [3], Zoran Arsic [2] and Rasa Mladenovic [3],*

[1] Department of Dentistry, Faculty of Medicine, University of Montenegro, 81000 Podgorica, Montenegro; mirko.mikic@t-com.me

[2] Department of Dentistry, Faculty of Medicine, University of Pristina, 38220 Kosovska Mitrovica, Serbia; zoran.vlahovic@t-com.me (Z.V.); zoran.arsic@med.pr.ac.rs (Z.A.)

[3] Department of Dentistry, Faculty of Medical Sciences, University of Kragujevac, 34000 Kragujevac, Serbia; momirstevanovic7@gmail.com

* Correspondence: rasa.mladenovic@med.pr.ac.rs; Tel.: +381-695302256

**Abstract:** This study aims to determine the correlation between the mean value of bone density measured on the CBCT device and the primary stability of dental implants determined by resonant frequency analysis. An experimental study was conducted on a material of animal origin: bovine femur and pig ribs. Two types of implants were used in this study: self-tapping and non-self-tapping of the same dimensions. Results of the experimental study showed a statistically significant correlation between bone density expressed in HU units and the primary stability of self-tapping and non-self-tapping dental implants expressed in ISQ units in bovine femur bones and self-tapping implants and pig rib bones. There was no statistically significant correlation between non-self-tapping dental implants in pig rib bones. Self-tapping and non-self-tapping implants did not show statistical significance in the primary stability in bones of different qualities. The analysis of bone density from CBCT images in the software of the apparatus expressed in HU units can be used to predict the degree of primary stability of self-tapping and non-self-tapping dental implants in bones of densities D1 and D2, and self-tapping dental implants in bones of the lower quality D4.

**Keywords:** bone density; CBCT; primary stability; osseointegration; dental implants

## 1. Introduction

Caries and periodontitis represent the most common causes of tooth loss. The replacement of the missing teeth is very important for the patient both from the health and from the psychosocial aspect [1–5].

Implant therapy, aimed at replacing the missing teeth, has been used successfully for the past 50 years and has also been recognized as an effective treatment option. The introduction of osseointegrated titanium implants back in 1965 resulted in the expansive development of implantology [6]. Numerous studies have shown that implant therapy is considered a predictable type of dental therapy with a very high average success rate of around 90–95% [7–11].

### 1.1. Osseointegration

The most important factor and prerequisite for achieving osseointegration, and thus the successful implant therapy, is the stability of the implant, which depends on the quality of the bone at the implant site, implant design and surgical implantation technique [12,13].

In order to achieve the appropriate conditions for osseointegration (or functional ankylosis), the implant must show proper initial fixation (primary stability) following the implantation [14].

The following factors determine the value of primary implant stability:

- quantity and quality of bone tissue at the implant site,
- implant design, and
- surgical implantation technique [15].

### 1.2. Alveolar Bone

Bone density stands as a significant predictor for the success of implant therapy [16]. Therefore, the evaluation of bone density represents an integral part of preimplantological clinical and radiographic examination.

Methods that enable a three-dimensional radiological presentation of the alveolar extensions of the upper and lower jaw include Computed Tomography (CT) and Cone Beam Computer Tomography (CBCT), which are the preferred methods for the analysis of bone density in the preimplantation phase [17].

Misch and Kircos [18] in 1999, and Northon and Gamble [19] in 2001 proposed a classification of bone density based on CT images using interactive software, with the data on bone quality at the site of the future implant being obtained based on the objective and quantitative result expressed in Hounsfield units.

### 1.3. Application of CBCT

Adequate radiological imaging during the planning process must provide quality imaging, and allow for realistic analysis and qualitative and quantitative measurements of the upper and lower jaw. Qualitative measurements, as well as bone density measurements, can be assessed and presented both visually and numerically using modern dental imaging Computed Tomography (CT) and Cone Beam Computer Tomography (CBCT) [20–23].

The principle of operation of the CBCT device is based on measuring the attenuation of X-rays that are absorbed differently by passing through different types of tissues. By passing through different tissues, the radiation weakens due to the absorption and scattering of X-rays. Detectors, after measurement, convert rays into electrical signals. The computer software synthesizes the image based on the data obtained from the detector. The synthesized image consists of the image matrix and its volume element (voxel) within which the pixel image element is created. A voxel is three-dimensional, and a pixel is two-dimensional.

In addition to using voxels of smaller dimensions to increase the accuracy of the HU number, algorithms are being developed that try to solve the problem of estimating the coefficient of linear attenuation for areas that are not fully recorded. [24]

CBCT imaging contains high spatial resolution images with voxel reconstructed CBCT data ranging between 0.07 and 0.4 mm [23]. Depending on the CBCT, an accuracy level of 200 µm should be feasible, however, with certain deviations [25].

The clinician has a software overview of the mean values of HU units in the given cylinder, i.e., in and around the virtually positioned implant, depending on the set parameters.

The advantages of CBCT over CT are reflected in the lower radiation to the patient during exposure, easier installation and lower cost of the device. Since its launch, CBCT has grown exponentially with over 85 different CBCT models being available [26].

In addition to the proven variations in the obtained HU values using CBCT and CT [27,28], there has been an increasing number of studies that use software analysis of bone density using CBCT to evaluate bone density [29–32].

### 1.4. Implant Design

As the design, shape and dimensions of implants can alter surgical outcomes (primary stability, bone compression) as well as biomechanical parameters (force distribution during occlusal function), various designs of commercially available implant systems have been developed with a view to providing optimal implant therapy to patients [33].

Implant macrodesign also applies to the shape and design of the thread, as well as the geometry, angle, slope, depth, thickness (width) and spacing of the thread. The most

important role of the macrodesign is to provide adequate stability after implantation, but also to ensure interaction with bone tissue through osseointegration [34,35].

### 1.5. Primary Implant Stability

The absence of clinical mobility of the implant following implantation represents the stability of the implant. Achieving and maintaining the stability of implants stands as a pre-requisite for successful osseointegration and the clinical outcome of dental implant therapy. The Resonance Frequency Analysis (RFA) method was first introduced in 1996 (Meredith) and is a non-invasive diagnostic method that enables clinical measurement of implant stability as well as the monitoring of the biological tissue response and osseointegration as a function of time [36,37].

With higher bone density (HU) values and a higher primary implant stability measured in ISQ units, Hausfield units can be used as a diagnostic parameter to assess possible implant stability. [38–40].

The models of the bovine rib and pork femur are analogous to human bone density, the software of the CBCT device on which the study was performed enables the virtual (guided) planning of the implant position that fully corresponds to the actual implant position built into the model in the next phase of the study. In this way, we obtained a direct relationship between the mean value of the bone density around the implant and the primary stability in ISQ units. The study was planned ex vivo because, in the following phases of the complete project, pathophysiological examinations of bone models were performed to examine the relationship between bone density and primary stability in more detail.

The authors considered the prevalence and accessibility of CBCT devices in clinical practice, whereby the therapists can assess bone density and select the appropriate implant in relation to the conditions, while utilizing adequate preimplant analysis and planning.

Clinical application is best reflected in the fact that the therapists may decide to use a non-tapping or more invasive self-tapping implant, or conduct additional procedures such as bone condensation or underprep drilling.

This study was performed under the hypothesis "Analysis of bone density of CBCT image (Cone Beam Computer Tomography) in the software of the device expressed in HU units (Hausfield Units—HU) can predict the value of primary implant stability, which stands as one of the basic factors for successful osseointegration, thus guiding the choice of the implant design".

The aims of the study:

- Determine the correlation between the mean value of the bone density measured on the CBCT device and the primary stability of self-tapping and non-self-tapping dental implants determined by resonant frequency analysis on samples of pig ribs and a bovine femur;
- Compare the obtained values of primary stability on self-tapping and non-self-tapping implants installed on samples of pig ribs and bovine femur samples.

## 2. Materials and Methods

This experimental study dealt with the correlation between radiological analysis of bone density and the primary stability of dental implants of different designs on animal origin material.

### 2.1. Experimental Animal Models

The experimental study used a bovine femur as a model of the human lower jaw (bone density D1/D2) and pork ribs of equal cortical thickness of 2 mm as a bone model of the human upper jaw (bone density D3/D4) [41].

All samples were obtained from experimental animals—males (due to higher bone density analogous to humans), six months old. Samples were provided from the local slaughterhouse. In order to preserve and minimize changes in the physical properties of the bone, the samples were prepared according to the instructions established by Sedline

and Hirch, which means that the bone was kept moist at all times, stored frozen in saline at −10 °C and used over the next 3–4 weeks [42].

For the purposes of the study, 20 samples of pork ribs and 20 samples of bovine femur were used.

### 2.2. Implants Used in the Study

In the experimental part of the study, two types of implants were used:

- self-tapping Bredent Narrow SKY (Bredent®, Weissenhorner Str. 2, 89250 Senden - Germany) dental implants, with the following dimensions: 3.5 × 10 mm, and
- non-self-tapping NobelReplace Conical Connection (Nobel Biocare, Nobel Biocare Services AG, P.O. Box, CH-8058 Zürich-Flughafen, Switzerland) with the following dimensions: 3.5 × 10 mm.

Implants had the same dimensions but different macro design of the threads. Both types of implants are recommended by the manufacturers for placement in bones of different quality.

The Bredent Narrow SKY features a conical, cylindrical implant shape, with double self-tapping compression threads [43]. NobelReplace Conical Connection is characterized by a conical shape, with non-aggressive, non-self-tapping threads [44,45].

In this experimental study, a total of 80 implants were used, that is 40 self-tapping and 40 non-tapping implants.

### 2.3. Individual Stent Fabrication

An individual stent or guide was made for each part of the rib and femur samples used in the study, using the appropriate material (3D Resin—Bredent, Germany) with two sleeves for the pilot drill with a diameter of 2.25 mm. Material for the experimental part of the study, bones of animal origin, were prepared and dimensionally adjusted. The prepared material, bones, were stamped with condensing silicones in order to obtain working models, cast from hard gypsum, used to make guides.

The guide was made of self-binding two-component acrylate 3D Resin—Bredent, Germany, which is thermally and dimensionally stable and recommended for laboratory development of guides in navigation implantology. Due to the release of heat during the bonding process, the making of the guide was done indirectly on a model, in order to avoid the possible influence of heat on the bone surface.

Preparation, fixation and marking of the sleeves was performed on the model. Then, each guide with sleeves was checked, marked and fixed to the bones (Figure 1).

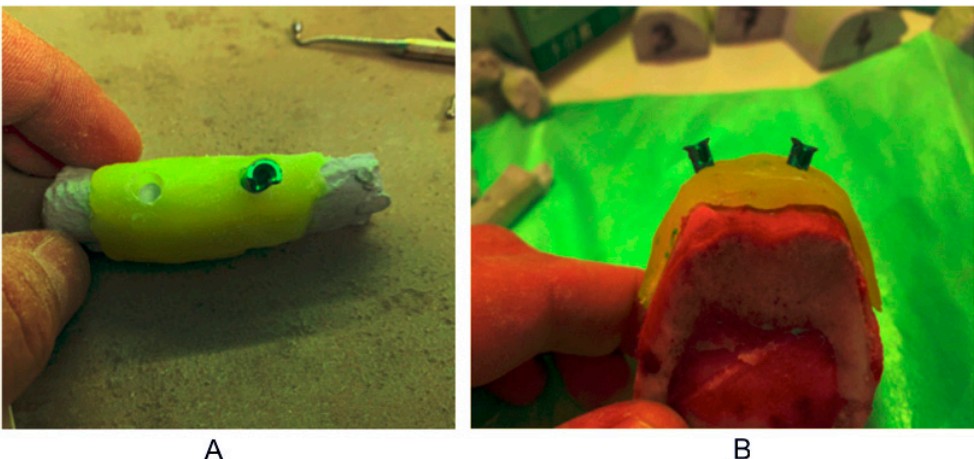

A                B

**Figure 1.** (**A**) Preparation and fixation of the sleeve; (**B**) test of a stent with sleeves.

### 2.4. Radiographic Analysis of Bone Density

Experimental material, parts of pig ribs and bovine femurs, with prepared, fixed and marked sleeves were recorded individually on a specially designed and adjusted stand of the CBCT device Planmeca 3D Promax—Asentajankatu 6, FI-00880 Helsinki, Finland.

All samples were recorded under the same conditions, 200 µm voxel, 90 kV, 10 mA, 36.4 s, 3112 mGy/cm$^2$. Multiplanar reconstruction was performed in Romexis software (Planmeca Romexis 5.3.4.39, Asentajankatu 6, FI-00880 Helsinki, Finland) with associated mathematical software algorithms for reducing CBCT artifacts. Prior to the start of the study, the calibration was performed by an authorized Planmeca support (Beam check, QA test and flat field calibration test).

According to the recommendations from the literature, we used smaller voxels (200 µm) and the latest version of Romexis software with accompanying algorithms to scatter scattered signs.

Three-dimensional reconstruction of the recorded samples was performed using the software Romexis (Planmeca Romexis 5.3.4.39, Asentajankatu 6, FI-00880 Helsinki, Finland) of the CBCT device.

In the software of the CBCT device, implants with sleeves, Bredent Narrow SKY and NobelReplace Conical Connection, dimensions 3.5 × 10 mm were selected from the implant base.

The selected implants are virtually placed so that the longitudinal axis of the implant coincides with the axis and shape of the sleeve.

After the virtual positioning of the implant, the program automatically produced the values of the average bone density, expressed in Hounsfield units in the cylinder, that is, inside the virtual implant and 1 mm in the surrounding bone. The mean value of bone density was used for the purposes of the research (Figure 2).

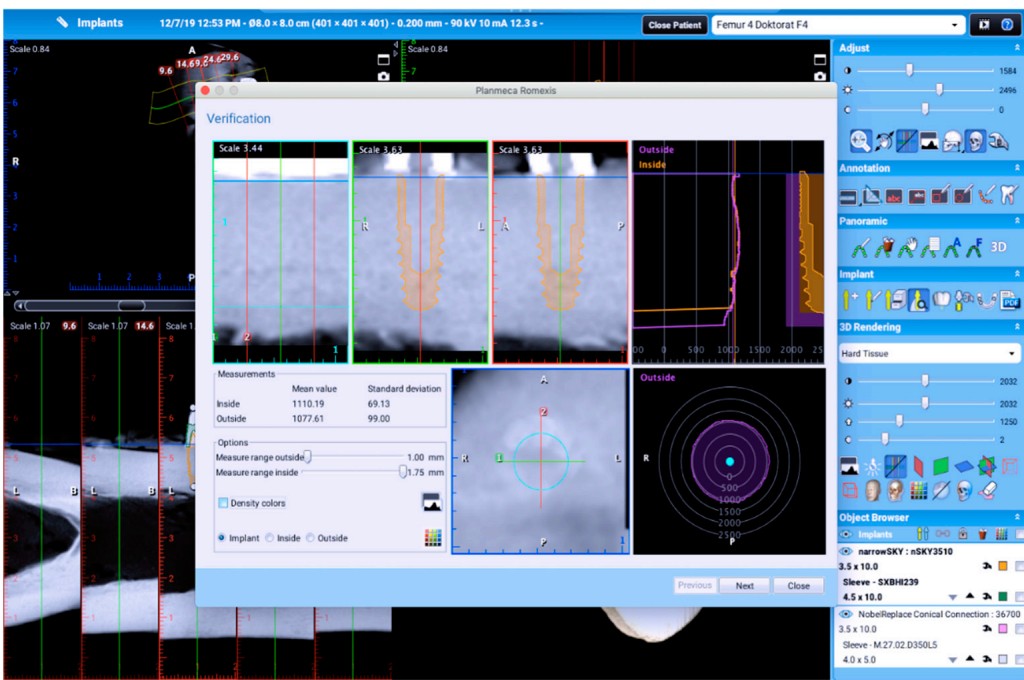

**Figure 2.** Software evaluations of bone volume expressed in HU.

### 2.5. Procedure for Experimental Implant Placement

After checking the position of the guide, the experimental bone was fixed and the experimental implant placement was initiated. The procedure involved preparation of the bearing in the bone and placement of the implant in the bearing, according to the

manufacturer's instructions, and specialized implant surgical sets Bredent® and Nobel Biocare.

The pilot drill was put through the sleeves from the navigation implantology set to a depth of 10 mm. This was followed by a preparation without the use of guides, using drills for the appropriate bone type and implant system (Figure 3).

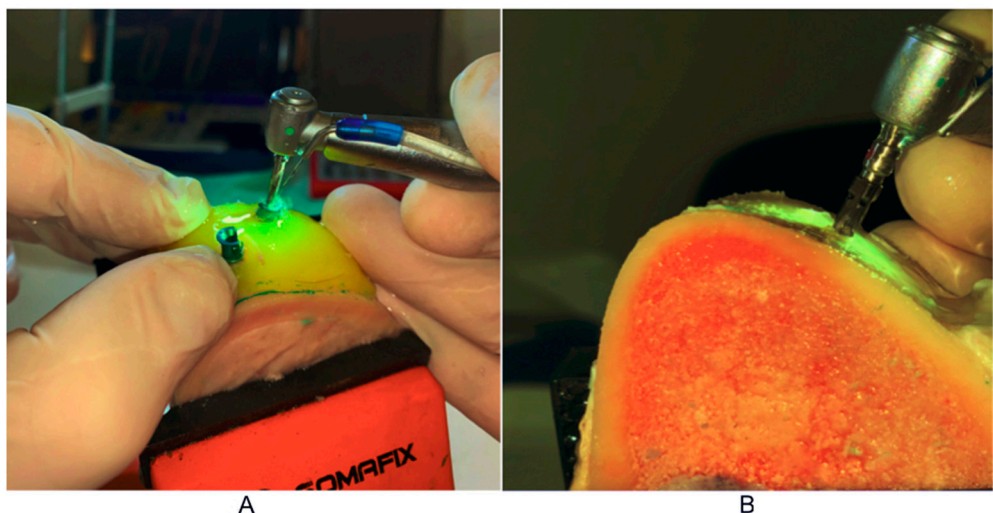

**Figure 3.** Bone preparation for implant bearing: (**A**) pilot drill through the sleeve; (**B**) preparation after the guide was removed.

One Bredent Narrow SKY self-tapping and one NobelReplace Conical Connection non-self-tapping implant were installed in each bovine femur and pig rib sample. The implants were placed in the bearing mechanically with a set torque of 35 N/cm$^2$.

### 2.6. Primary Stability Measurement Procedure

After placement of both implants, measurements of primary implant stability were performed using resonant frequency analysis (RFA), via Osstell mentor device (Integration Diagnostics AB, Stampgatan 14, 411 01 Göteborg, Sweden).

The implant is placed and tightened manually, using a force of 4 to 5 Ncm and a suitable SmartPeg, type 49 for Bredent Narrow SKY and type 60 for NobelReplace Conical Connection.

The Osstell mentor probe is placed at the right angle and with a distance of 2–3 mm, next to the SmartPeg, in four positions: buccal, oral, mesial and distal. There are two threads on the top of the device. After switching on, the first thread becomes a magnet that excites the magnet on the Smartpeg. The second thread registers the vibration produced by Smartpeg. After a short sound, the ISQ value is read on the register of the device. During this study, we used the mean ISQ value obtained from four different directions.

### 2.7. Statistical Analysis

Descriptive statistical methods, methods for testing statistical hypotheses, methods for testing correlations, and methods for examining the correlation between outcomes and potential predictors were used to analyze primary data. Depending on the type of variables, the data description is given as *n* (%) or as ±sd.

*T*-test was used as a method for testing statistical hypotheses. The Pearson linear correlation coefficient was used to examine the correlation of the two variables.

Statistical hypotheses were tested at the level of statistical significance of 0.05.

The obtained data were then statistically processed to obtain a correlation between the mean value of bone volume density and the value of primary stability of implants.

The following measurements were performed during the study:

- Bone densities based on CBCT images in HU units;
- Primary stability of dental implants in ISQ units.

## 3. Results

An experimental study on the material of animal origin was conducted using 20 samples of pig ribs and 20 samples of bovine femurs, in which two implants were installed. In this study, a total of 40 self-tapping implants and 40 non-self-tapping implants (50%) were installed.

Table 1 shows the distribution of mean values of the bone density expressed in Hounsfield units in relation to the bovine femur and pork rib.

**Table 1.** Overview of bone density values expressed in HU units relative to the bone.

| HU | *n* | $\bar{x}$ | sd | Med | Min | Max |
|---|---|---|---|---|---|---|
| Bovine femur | 40 | 851.8 | 193.0 | 827.2 | 422.9 | 1236.9 |
| Pig rib | 40 | 255.7 | 66.1 | 254.1 | 99.7 | 388.6 |

The arithmetic mean and standard deviation of bone density expressed in HU units in the bovine femur stood at $851.8 \pm 193.0$, while in the pig rib it was $255.7 \pm 66.1$, which is a statistically significant difference (t = 18,478; *p* < 0.001).

Figure 4 shows the correlation between the mean values of bone density measured on a CBCT device and expressed in HU units and the primary stability of self-tapping dental implants installed in a pig rib.

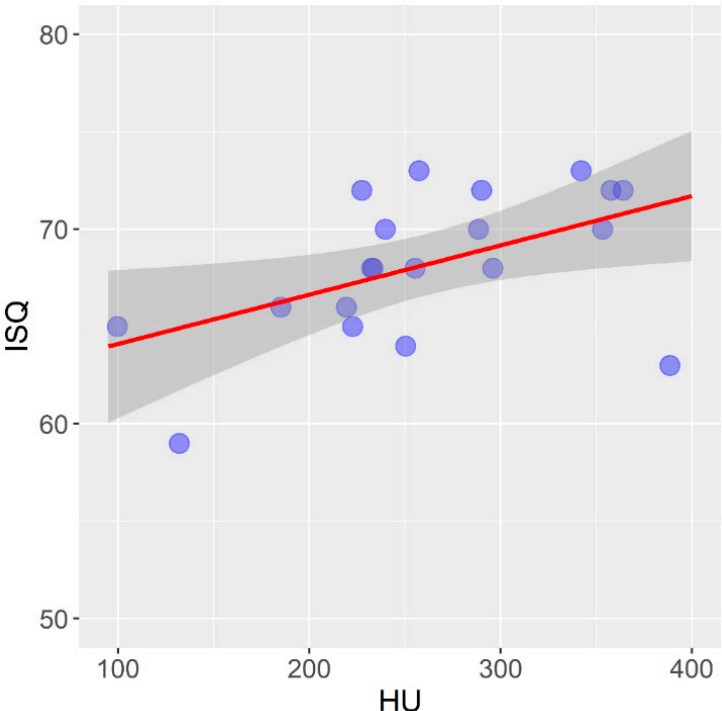

**Figure 4.** Correlation between bone density and primary stability of self-tapping implants in pig ribs.

In self-tapping implants in the pig rib, there is a statistically significant mean positive correlation between the bone density expressed in HU units and the primary stability of the dental implants expressed in ISQ units (r = 0.506; *p* = 0.023).

Figure 5 shows the correlation between the mean values of bone density measured on a CBCT device and expressed in HU units and the primary stability of non-self-tapping dental implants in pig ribs.

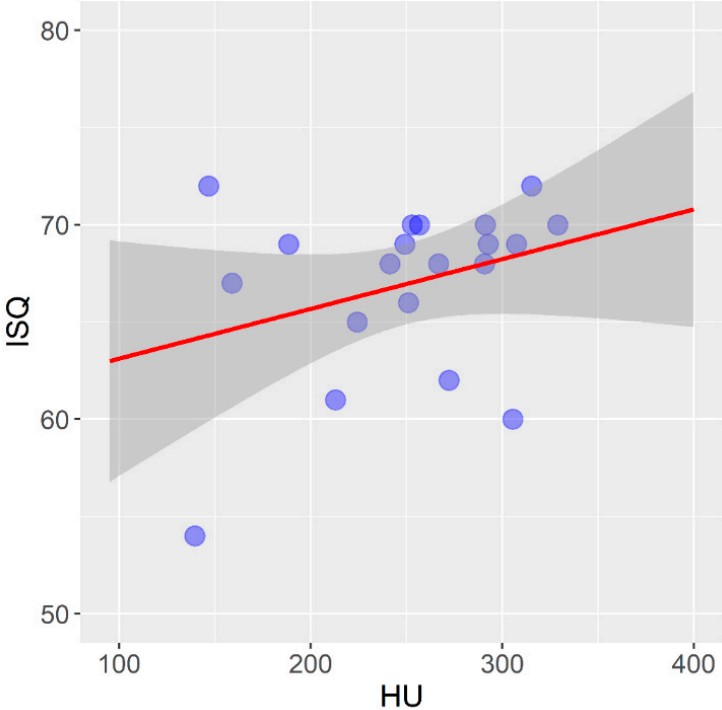

**Figure 5.** Correlation between bone density and primary stability of non-self-tapping implants in pig ribs.

In non-self-tapping implants in the pig rib, there is no statistically significant correlation between the bone density, expressed in HU units, and the primary stability of the dental implants, expressed in ISQ units ($r = 0.318$; $p = 0.172$).

Table 2 shows the values of primary stability of the self-tapping and non-self-tapping implants in the pig rib.

**Table 2.** ISQ value in pig rib in relation to implant type.

| ISQ | $n$ | $\bar{x}$ | sd | Med | Min | Max |
|---|---|---|---|---|---|---|
| Self-tapping | 20 | 68.2 | 3.8 | 68.0 | 59.0 | 73.0 |
| Non-self-tapping | 20 | 67.0 | 4.5 | 68.5 | 54.0 | 72.0 |

The arithmetic mean and standard deviation of the values of the primary stability of the self-tapping implants expressed in ISQ units in the pig rib was $68.2 \pm 3.8$, while in the non-self-tapping implants it was $67.0 \pm 4.5$, which is not a statistically significant difference ($t = 0.947$; $p = 0.350$).

Mean bone density values measured on the CBCT device, expressed in HU units, and their correlation to the primary stability of the self-tapping dental implants installed in the bovine femur are given in Figure 6.

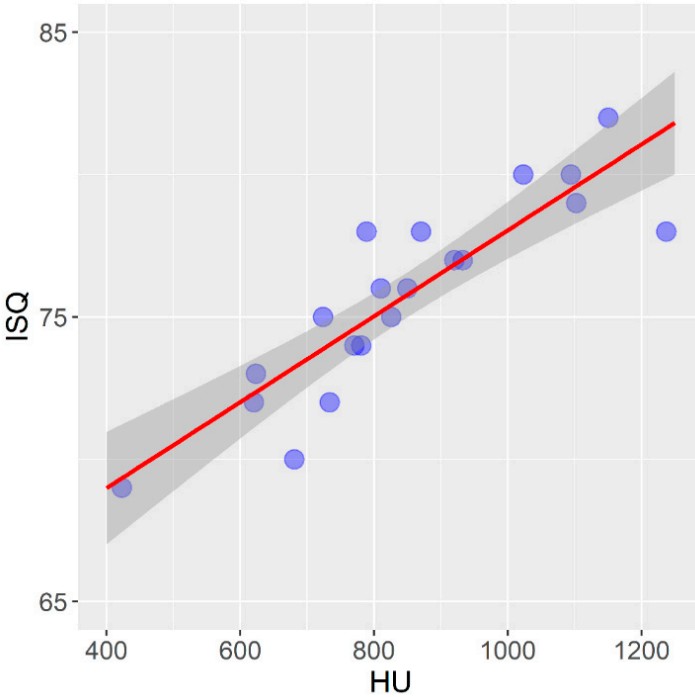

**Figure 6.** Correlation between bone density and primary stability of self-tapping implants in bovine femur.

In the self-tapping implants installed in the bovine femur, there is a statistically significant strong positive correlation between the bone density expressed in HU units and the primary implant stability expressed in ISQ units ($r = 0.880$; $p < 0.001$).

Figure 7 shows the correlation between the mean values of the bone density measured using a CBCT device and expressed in HU units and the primary stability of the non-self-tapping dental implants in the bovine femur.

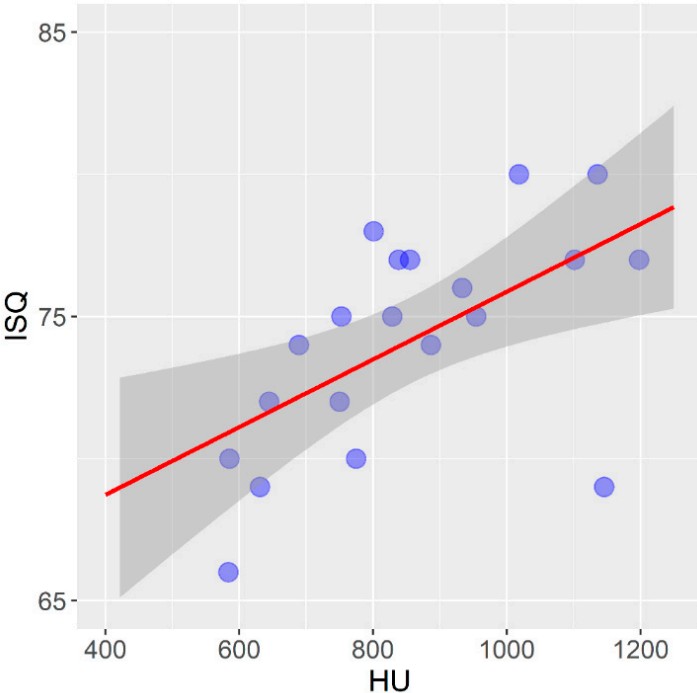

**Figure 7.** Correlation between bone density and primary stability of non-self-tapping implants in bovine femur.

In the non-self-tapping implants in the bovine femur, there is a statistically significant mean positive correlation between the bone density expressed in HU units and the primary implant stability expressed in ISQ units (r = 0.584; $p$ = 0.007).

The values of the primary stability of the self-tapping and non-self-tapping implants in the bovine femur are shown in Table 3.

**Table 3.** The value of ISQ in bovine femur with respect to the implant type.

| ISQ | $n$ | $\bar{x}$ | sd | Med | Min | Max |
|---|---|---|---|---|---|---|
| Self-tapping | 20 | 75.8 | 3.4 | 76.0 | 69.0 | 82.0 |
| Non-self-tapping | 20 | 74.2 | 3.9 | 75.0 | 66.0 | 80.0 |

The arithmetic mean and standard deviation of the primary stability of the self-tapping implants in the bovine femur was 75.8 ± 3.4 ISQ, while in the non-self-tapping implants it was 74.2 ± 3.9 ISQ, which does not represent a statistically significant difference (t = 1381; $p$ = 0.175).

## 4. Discussion

In today's modern dental therapy, implant therapy represents the therapy of choice, indicated for all types of toothlessness. With the development of implantology and through the use of a multidisciplinary medical approach, most contraindications for implant therapy have been translated from absolute to relative contraindications, with a tendency to reduce complications and increase the success rate. The success rate of implant therapy, according to most studies, ranges from 90% to 95%.

There are numerous factors that influence the success of implant therapy, and are usually related to the characteristics of the patient, the type of implant and the skills of the therapist [46–52].

The experimental part of this study, which was performed on the material of animal origin, pig ribs and a bovine femur, assessed the influence of bone density, based on CBCT images, on the primary stability in different designs of dental implants, self-tapping and non-self-tapping.

The expansive development of radiological diagnostics and its application in implantology has enabled objective pre-implant preparation as well as the selection of the optimal treatment plan. Qualitative and objective bone analysis was enabled with the introduction of a three-dimensional multiplanar overview of the jaw bone tegmen, first by using Computed Tomography (CT), then by using Multilayer Computed Tomography (MSCT) and by applying Cone Computed Tomography (CBCT) to implant practices [53,54].

The evaluation of the quality of the bone density in both the experimental part of the study on bones of animal origin, as well as in the clinical part of the study on humans, was performed on the basis of the software analysis of CBCT images and expressed in HU units. The validity of the assessment of the bone density expressed in HU units, based on CBCT and CT images has been the focus of numerous studies. Armstrong (2006) and Arisan et al. (2013) pointed out that there is a difference in the values of HU units in the analysis of material recorded under the same conditions on CT and CBCT [54,55]. The research for the assessment of the bone density, expressed in HU units, can include the analysis of CT images and the analysis of CBCT images as valid methods for the evaluation of bone density and is the method of choice in the preimplantology phase. Due to its availability, lower radiation dose, and simpler installation, CBCT represents a more common method in the dental practice [53,56–62].

In the experimental study, the average bone density expressed in HU units in the bovine femur, which was used as a human lower jaw model, was 851.8 HU. The lowest obtained average value of the bovine femur bone density amounted to 442.9 HU, while the highest was 1236.9 HU. All of the obtained values correspond to the bone quality classified in categories D1 and D2, according to Misch [23], and Q1, Q2 and Q3 according to Norton

and Gamble [24], which, according to these authors, can be found locally in the alveolar extensions of the lower jaw.

The average bone density in the pig rib, which was used as a model of the human upper jaw, was 255.7 HU. The minimum value was 99.7 HU and the maximum 388.6 HU. The average value of the pork rib bone quality corresponds to the bone quality classified in category D4 according to Misch [18] and Q4 according to Norton and Gamble [19], which, according to these authors, can be found in the lateral region of the upper jaw. The partial volume effect, as described in the literature, can explain the registered minimum value of the average bone density on one sample of pork rib without a noticeable deviation of the primary stability and a macroscopically noticeable lower strength on that sample. When different types of tissues are found in one voxel, in this case, soft tissue cavities and bone beams, with different X-ray attenuation coefficients, this sometimes affects the image quality and may result in an inaccurate CT or CBCT reading [57].

The results of the presented experimental study did not show a significant statistical difference in the primary stability between the self-tapping and non-self-tapping implants in both types of bone tissue. In an in vitro study aimed at examining the correlation of the primary stability of two types of implants in different types of bone tissue, Bilhan et al. came to similar results, and that is that there is no significant statistical difference (2015). The results of the values of the primary stability in their study are slightly higher for both types of implants compared to our study, which can be explained by the use of implants of a larger diameter and length [58]. Falco et al. (2018), whose study was also performed on a material of animal origin, showed a statistically significant difference in the primary stability in the self-tapping implants compared to non-self-tapping implants in a low-density D4 bone, while in other bone types there was no significant difference [59]. Other studies dealing with the primary stability of implants of different designs conclude that the main determinants of primary stability are the bone density and implant macrodesign, and that self-tapping implants are recommended for lower-density bones and for immediate implantation [60–64].

According to the results of the presented experimental study on pig ribs, there is a statistically significant mean positive correlation between the bone density expressed in HU units and the primary stability of self-tapping dental implants expressed in ISQ units (r = 0.506; *p* = 0.023). While there was no statistically significant difference in this type of bone when it comes to non-self-tapping dental implants, Isoda et al. (2012), in a similar in vitro study using a material of animal origin, demonstrated a significant positive correlation between the bone density measured on a CBCT device and the primary stability of self-tapping dental implants [65]. Möhlhenrich et al. (2019) published similar results on the correlation between bone density measured on the CBCT device and the primary implant stability [66]. The difference in the correlation between the bone density assessment and primary stability in the two types of implants in the lower-density bone can be explained by the macrodesign of the thread. In their review paper, when analyzing thirteen papers that analyzed the correlation between different factors and the primary stability of implants, Marquezan et al. (2012) concluded that self-tapping implants with more aggressive threads show a better primary stability compared to non-self-tapping dental implants [67].

In self-tapping implants installed in the bovine femur, there is a statistically significant strong, and in non-self-tapping, a moderately strong positive correlation between the bone density expressed in HU units and the primary implant stability expressed in ISQ units in the presented study. Fuster-Torres et al. (2001) and Isoda et al. (2012) reported a positive correlation between the bone density expressed in HU units and the primary implant stability, which coincides with the results of the presented study [65,68].

## 5. Conclusions

On the basis of the presented results, we can conclude the following:

- By analyzing the density of the bone tissue in the CBCT images in the software of the device expressed in HU units, we cannot predict the degree of the primary stability

of the non-self-tapping dental implants in bones of a lower quality D4, according to Misch, and Q4, according to Norton and Gamble;

- Self-tapping and non-self-tapping dental implants installed in D4- and Q4-quality bones do not show a significant statistical difference in the primary stability;
- By analyzing the density of the bone tissue in the CBCT images in the software of the device expressed in Hausfield units, we can predict the degree of the primary stability of the self-tapping dental implants in bones of the densities D1, D2 and Q1–Q3;
- By analyzing the density of the bone tissue in the CBCT images in the software of the device expressed in HU units, we can predict the degree of the primary stability of the non-self-tapping dental implants in bones of the densities D1, D2 and Q1–Q3.

**Author Contributions:** Writing—review and editing, M.M.; Conceptualization, Z.V.; methodology, Z.A. and M.S.; Supervision, R.M. All authors have read and agreed to the published version of the manuscript.

**Funding:** This research received no external funding.

**Institutional Review Board Statement:** The research was approved by the Ethics Committee of the Clinical Centre of Montenegro no. 03/01-4536/1 from March 2018.

**Informed Consent Statement:** Not applicable.

**Data Availability Statement:** Data available on request from authors.

**Conflicts of Interest:** The authors declare no conflict of interest.

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
