# Peer review of "The Importance of Correlation between CBCT Analysis of Bone Density and Primary Stability When Choosing the Design of Dental Implants—Ex Vivo Study"

_tomography, doi:10.3390/tomography8030107_

Round 1
Reviewer 1 Report
Dear authors,
your study is interesting and well-conducted. Congratulations on the work done.
I suggest revising the text to remove some typos, improve your English and make it clearer and more understandable.
Detailed comments are as follows:
1. What is the main question addressed by the research?
The aim of this study is to analyze the bone density of CBCT 159 image with software to evaluate the predictability of the value of primary implant stability
2. Do you consider the topic original or relevant in the field, and if so, why?
Primary stability is one of the most important elements of implant success. Being able to define through CBCT the keys to success and failure of implant placement is of fundamental importance. The originality of the study is present both in the interesting objective and in the method used, also exploiting a good variety of implant designs
3. What does it need to add to the subject area compared with other published material?
It might be useful to insert and detail the applicability of the clinical implications of the research and to discuss more appropriately the limits of "ex vivo" work and how an "in vivo" protocol, even if retrospective, could be proposed.
4. What specific improvements could the authors consider regarding the methodology?
An evaluation of the error in the measurements could be added. I would recommend repeating the measurements performed by the same operator and having others run for one second to evaluate the intra and inter operator error.
5. Are the conclusions consistent with the evidence and arguments presented and do they address the main question posed?
The conclusions are well supported by the results. They argue well about the intent of the work and well define what emerged from the research.
6. Apart from the problem of English expression, what are the main shortcomings of the manuscript?
The limit of the study is certainly that of working on ex vivo samples and on a defined number of implant morphologies that certainly do not reflect all the products on the market.
7. Are the references appropriate?
Please consider adding these references to the manuscript:
Dental implant primary stability in different regions of the Jawbone: CBCT-based 3D finite element analysis. Saudi Dent J. 2020 Feb;32(2):101-107. doi: 10.1016/j.sdentj.2019.06.001. Epub 2019 Jun 17. PMID: 32071539; PMCID: PMC7016247.
Evaluating cortico-cancellous ratio using virtual implant planning and its relation with the immediate and long-term stability of a dental implant- A CBCT-assisted prospective observational clinical study. Niger J Clin Pract. 2019 Jul;22(7):982-987. doi: 10.4103/njcp.njcp_22_19. PMID: 31293265.
A Retrospective Study on Insertion Torque and Implant Stability Quotient (ISQ) as Stability Parameters for Immediate Loading of Implants in Fresh Extraction Sockets. Biomed Res Int. 2019 Nov 3;2019:9720419. doi: 10.1155/2019/9720419. PMID: 31781659; PMCID: PMC6875416.
Author Response
Thanks for the comments. The attached file contains the answers to your comments.

Reviewer 2 Report
Dear authors congratulations for the manuscript, in my opinion it is a well-designed and conducted study. however, before publications some minor modifications are suggested to improve readibility.
- ABSTRACT: contains information duplicated , please try to not duplicate information in the abstract as the obective and the materials and metods contain almost the same info
- INTRODUCTION: The introduction is well ordered and with enough information however the distribution of the different paragraphs (1.1, 1.2,1.3.. ) does not make sense, please avoid using this or number it in a different way
- LINE 146. Please provide a reference to confirme the affirmation regarding similarities between human and pigs regarding bone density
- LINE 165: Please summarize the objective and avoid writing al the secondary objectives this way
- 182:duplication information
- LINE 244, avoid using personal pronouns , scientific language use impersonal and passive forms
- try to summarize figure, so many figures confuse the readears
- references must be double checked as the are not consistent in form
- I would appreciate if more emphasis is marked about the clinical relevance of the study
Congratulations again for the study, regards
Author Response

(The authors gave the same response as above.)

Reviewer 3 Report
Dear Authors,
you made a great work! However, some little changes are mandatory before acceptance.

Author Response

(The authors gave the same response as above.)
